# Out-of-equilibrium dynamical properties of Bose-Einstein condensates in a ramped-up weak disorder

Rodrigo P. A. Lima[1,2⋆], Milan Radonjić[3,4†] and Axel Pelster[5‡]

**1** GISC & InterTEP, Departamento de Física, Facultad de Ciencias Ambientales y Bioquímica, Universidad de Castilla-La Mancha, 45071 Toledo, Spain
**2** GFTC, Instituto de Física, Universidade Federal de Alagoas, Maceió AL 57072-970, Brazil
**3** I. Institut für Theoretische Physik, Universität Hamburg, Notkestraße 9, 22607 Hamburg, Germany
**4** Institute of Physics Belgrade, University of Belgrade, Pregrevica 118, 11080 Belgrade, Serbia
**5** Fachbereich Physik und Forschungszentrum OPTIMAS, Rheinland-Pfälzische Technische Universität Kaiserslautern-Landau, Erwin-Schrödinger Straße 46, 67663 Kaiserslautern, Germany
⋆ rodrigo.paula@uclm.es, † milan.radonjic@uni-hamburg.de, ‡ axel.pelster@rptu.de

November 26, 2025

## Abstract

We theoretically study how the superfluid and condensate deformation of a weakly interacting ultracold Bose gas evolve during the ramp-up of an external weak disorder potential. Both resulting deformations turn out to consist of two distinct contributions, namely a reversible equilibrium one, already predicted by Huang and Meng in 1992, and a nonequilibrium dynamical one, whose magnitude depends on the details of the ramping protocol. For the specific case of the exponential ramp-up protocol, we are able to derive analytical time-dependent expressions for the above quantities. After a sufficiently long time, a steady state emerges that is generically out of equilibrium. We take the first step in investigating its properties by studying its relaxation dynamics. In addition, we analyze the two-time correlation function and elucidate its relation to the equilibrium and the dynamical part of the condensate deformation.

# 1   Introduction

The concept of disorder, in the form of a frozen random potential landscape, allows us to realistically describe the distribution of impurities, defects, and other types of imperfections in various quantum many-body systems, ranging from solid-state materials to ultracold quantum gases. It is generally assumed that disorder is an unavoidable nuisance that weakens or even prevents the emergence of quantum effects. However, under certain circumstances, disorder has turned out to be crucial for novel quantum phenomena that have no clean counterpart [1]. Prime examples are Anderson localization of one-particle wave functions [2–5] or many-body localization in isolated many-body systems [6–8]. These successes have led to the idea that one can use properly tailored disorder as a tuning knob for open-system control of quantum many-body systems.

However, for many disordered systems that occur in nature, the random potential landscape is not frozen, but changes in time either deterministically or stochastically. Thus, it becomes important to study the disorder from the broader perspective that it may change on some time scale. Such changes may occur naturally in an experimental situation when the perturbation is turned on and off. Dynamic disorder with tunable correlation time [9] or quenching [10] is often used intentionally to probe the properties of quantum many-body systems. For example, quenching of disorder in an ultracold bosonic gas in a lattice has been used experimentally to dynamically probe the quantum phase transition of the superfluid-Bose glass at non-zero temperature [11]. Moreover, quenching can also lead to the appearance of non-trivial steady states, as shown, for example, by the theoretical study of an interaction quench of a 3D BEC in a static disordered potential [12]. There it was found that the quench dynamics enhances the ability of the disorder to deplete the superfluid more than to deform the condensate. In Ref. [13] we found that after the disorder is ramped up, the resulting stationary condensate deformation turns out to be a sum of two parts. One is an equilibrium part, which actually corresponds to the adiabatic switching on of the disorder and was already found by Huang and Meng in 1992 [14]. The other represents a dynamically induced part, which depends on how fast the disorder is turned on.

The latter peculiar discovery warrants a more detailed investigation, which is carried out in the present paper. In Sec. 2 we present the underlying perturbative mean-field theory for describing a homogeneous Bose gas moving in a temporally controlled weak disorder potential. The subsequent Sections apply the formalism to the exponential ramp-up protocol. We determine analytical, time-dependent expressions for the superfluid deformation in Sec. 3 and for the condensate deformation in Sec. 4. A detailed analysis corresponding to the correlation functions is given in Sec. 5, followed by concluding remarks in Sec. 6. The applied contour integration method is outlined in Appendix A.

## 2 Perturbative mean-field theory

At the initial time $t = 0$, a Bose-Einstein condensate of identical bosons $N$ at zero temperature occupies the volume $V$ and has a constant particle density $n = N/V$. It moves with velocity $\mathbf{v} = \hbar\mathbf{K}/m$ and is described by the homogeneous wave function $\exp(-i\mu_0 t/\hbar)\Psi_0(\mathbf{x}, t)$, where

$$\Psi_0(\mathbf{x}, t) = \exp\left(i\mathbf{K}\cdot\mathbf{x} - i\frac{\hbar K^2}{2m}t\right)\sqrt{n}\,, \tag{1}$$

and $\mu_0 = gn$ is the equilibrium chemical potential, with $g$ being the contact interaction strength. The thermodynamic limit $N, V \to \infty$ with constant $n$ will be implicitly assumed in the final stage of all calculations. The condensate dynamics is modeled by the mean-field time-dependent Gross-Pitaevskii equation, which at the time $t = 0$ reads

$$i\hbar\frac{\partial\Psi_0(\mathbf{x}, t)}{\partial t} = \left[-\frac{\hbar^2\nabla^2}{2m} - \mu_0 + g|\Psi_0(\mathbf{x}, t)|^2\right]\Psi_0(\mathbf{x}, t)\,. \tag{2}$$

The important characteristic time scale of mean-field dynamics is determined by the inverse of the chemical potential $\tau_{\mathrm{MF}} \equiv \hbar/\mu_0$. Physically, it establishes the order of magnitude related to the ratio of the condensate healing length and the speed of sound.

At later times $t > 0$, a weak external disorder potential $u(\mathbf{x})$ is ramped up via the drive function $f(t)$, which takes values between 0 and 1, so that $f(0) = 0$ and $\lim_{t\to\infty} f(t) = 1$. We assume that the disorder potential has zero ensemble average at every point, $\langle u(\mathbf{x})\rangle = 0$, in order to eliminate the effects of a simple shift of the chemical potential. The two-point correlation function is of the form

$$\langle u(\mathbf{x})u(\mathbf{x}')\rangle = \mathcal{R}(\mathbf{x} - \mathbf{x}')\,, \tag{3}$$

so that homogeneity is restored after the disorder ensemble averaging is performed. In the $\mathbf{k}$-space we have, correspondingly,

$$\langle\tilde{u}(\mathbf{k})\rangle = 0\,, \quad \langle\tilde{u}(\mathbf{k})\tilde{u}(\mathbf{k}')\rangle = (2\pi)^3\delta(\mathbf{k} + \mathbf{k}')\tilde{\mathcal{R}}(\mathbf{k})\,, \tag{4}$$

where, from now on, the tilde will denote the spatial Fourier transform. The evolution of the condensate is now described by the disordered wave function $\Psi(\mathbf{x}, t)$ obeying the time-dependent Gross-Pitaevskii equation [13, 15], which includes the time-dependent random potential

$$i\hbar\frac{\partial\Psi(\mathbf{x}, t)}{\partial t} = \left[-\frac{\hbar^2\nabla^2}{2m} + u(\mathbf{x})f(t) - \mu_0 + g|\Psi(\mathbf{x}, t)|^2\right]\Psi(\mathbf{x}, t)\,. \tag{5}$$

For convenience, we introduce the auxiliary wave function $\psi(\mathbf{x}, t)$ via the relation

$$\Psi(\mathbf{x}, t) = \exp\left(i\mathbf{K}\cdot\mathbf{x} - i\frac{\hbar K^2}{2m}t\right)\psi(\mathbf{x}, t)\,. \tag{6}$$

Note that due to the restored homogeneity, one has $\nabla\langle\psi(\mathbf{x}, t)\rangle = 0$. We will consider the regime where the disorder is a perturbation that is small compared to all other energy scales. In this case, the auxiliary wave function $\psi(\mathbf{x}, t)$ can be expanded as

$$\psi(\mathbf{x}, t) = \psi_0 + \psi_1(\mathbf{x}, t) + \psi_2(\mathbf{x}, t) + \ldots\,, \tag{7}$$

where $\psi_0 = \sqrt{n}$, while $\psi_\alpha(\mathbf{x}, t = 0) = 0$ and $\psi_\alpha(\mathbf{x}, t) = \mathcal{O}(|u(\mathbf{x})|^\alpha)$ for $\alpha \geq 1$ are perturbative corrections due to the disorder, which will allow us to calculate various quantities of interest.

The disorder ramp-up with the condensate at rest, i.e. $\mathbf{K} = \mathbf{0}$, was analyzed in [13]. There, the primary quantity of interest was the condensate deformation [16], which up to the second order is

$$q(t) = \langle |\Psi(\mathbf{x}, t)|^2 \rangle - |\langle \Psi(\mathbf{x}, t) \rangle|^2 \approx \langle |\psi_1(\mathbf{x}, t)|^2 \rangle - |\langle \psi_1(\mathbf{x}, t) \rangle|^2. \tag{8}$$

As shown in [13], it represents a hallmark of non-equilibrium steady states of the system reached long after the disorder has been ramped up. However, the observed non-equilibrium steady states have not been characterized. The first step towards such a goal is to study the dynamics and steady-state values of additional system quantities. With this in mind, we investigate here the superfluid properties of the disordered condensate. The disorder-averaged momentum density can be decomposed as

$$\langle \mathbf{p}(t) \rangle \equiv -i\hbar \langle \Psi^*(\mathbf{x}, t) \nabla \Psi(\mathbf{x}, t) \rangle = \hbar \mathbf{K} \langle |\Psi(\mathbf{x}, t)|^2 \rangle - i\hbar \langle \psi^*(\mathbf{x}, t) \nabla \psi(\mathbf{x}, t) \rangle. \tag{9}$$

Since $\langle |\Psi(\mathbf{x}, t)|^2 \rangle = n$, the first term represents the momentum density of a clean homogeneous moving condensate and we introduce the momentum density deformation as

$$\langle \delta \mathbf{p}(t) \rangle = i\hbar \langle \psi^*(\mathbf{x}, t) \nabla \psi(\mathbf{x}, t) \rangle \approx i\hbar \langle \psi_1^*(\mathbf{x}, t) \nabla \psi_1(\mathbf{x}, t) \rangle, \tag{10}$$

where the last expression is the second-order approximation in terms of first-order corrections. The momentum density gives us access to the superfluid density tensor [17, 18] through the relation

$$n_{s,ij}(t) = \frac{1}{\hbar} \frac{\partial \langle p_i(t) \rangle}{\partial K_j} \bigg|_{\mathbf{K}=\mathbf{0}}. \tag{11}$$

By analogy, we define the superfluid deformation tensor in the presence of disorder as

$$\delta n_{s,ij}(t) = \frac{1}{\hbar} \frac{\partial \langle \delta p_i(t) \rangle}{\partial K_j} \bigg|_{\mathbf{K}=\mathbf{0}}, \tag{12}$$

which obeys the relation $n_{s,ij}(t) = n\delta_{ij} - \delta n_{s,ij}(t)$.

Furthermore, we are interested in the long-time behavior of the two-point connected correlation function

$$\langle \psi(\mathbf{x}, t) \psi^*(\mathbf{y}, t + T) \rangle_c = \langle \psi(\mathbf{x}, t) \psi^*(\mathbf{y}, t + T) \rangle - \langle \psi(\mathbf{x}, t) \rangle \langle \psi^*(\mathbf{y}, t + T) \rangle, \tag{13}$$

where $T$ is the time delay, and relate its various limits to specific parts of the condensate depletion. Using the second-order result for the correlation function

$$\begin{aligned} \langle \psi(\mathbf{x}, t) \psi^*(\mathbf{y}, t + T) \rangle \approx {}& \psi_0^2 + \psi_0 \langle \psi_1(\mathbf{x}, t) + \psi_1^*(\mathbf{y}, t + T) \rangle + \langle \psi_1(\mathbf{x}, t) \psi_1^*(\mathbf{y}, t + T) \rangle \\ & + \psi_0 \langle \psi_2(\mathbf{x}, t) + \psi_2^*(\mathbf{y}, t + T) \rangle, \end{aligned} \tag{14}$$

as well as

$$\begin{aligned} \langle \psi(\mathbf{x}, t) \rangle \langle \psi^*(\mathbf{y}, t + T) \rangle \approx {}& \psi_0^2 + \psi_0 \langle \psi_1(\mathbf{x}, t) + \psi_1^*(\mathbf{y}, t + T) \rangle + \langle \psi_1(\mathbf{x}, t) \rangle \langle \psi_1^*(\mathbf{y}, t + T) \rangle \\ & + \psi_0 \langle \psi_2(\mathbf{x}, t) + \psi_2^*(\mathbf{y}, t + T) \rangle, \end{aligned} \tag{15}$$

we find that the connected correlation function up to the second order depends only on the first-order wave function corrections

$$\langle \psi(\mathbf{x}, t) \psi^*(\mathbf{y}, t + T) \rangle_c \approx \langle \psi_1(\mathbf{x}, t) \psi_1^*(\mathbf{y}, t + T) \rangle - \langle \psi_1(\mathbf{x}, t) \rangle \langle \psi_1^*(\mathbf{y}, t + T) \rangle. \tag{16}$$

To gain access to the second-order correct quantities of interest, we need only determine the first-order perturbative corrections using Eq. (5). Since the resulting equations for the corrections are linear, we use the Fourier and Laplace transformations, which reduce the problem to an algebraic linear system

$$i\hbar s \tilde{\psi}_1(\mathbf{k}, s) = \left[ \hbar\omega_k + \frac{\hbar^2}{m}\mathbf{K}\cdot\mathbf{k} + g\psi_0^2 \right] \tilde{\psi}_1(\mathbf{k}, s) + g\psi_0^2 \tilde{\psi}_1^*(\mathbf{k}, s) + \tilde{u}(\mathbf{k})f(s)\psi_0, \quad (17a)$$

$$-i\hbar s \tilde{\psi}_1^*(\mathbf{k}, s) = \left[ \hbar\omega_k - \frac{\hbar^2}{m}\mathbf{K}\cdot\mathbf{k} + g\psi_0^2 \right] \tilde{\psi}_1^*(\mathbf{k}, s) + g\psi_0^2 \tilde{\psi}_1(\mathbf{k}, s) + \tilde{u}(\mathbf{k})f(s)\psi_0, \quad (17b)$$

where $\hbar\omega_k = \hbar^2 k^2/(2m)$ is the free particle dispersion. This linear system has the non-trivial solution

$$\tilde{\psi}_1(\mathbf{k}, s) = -\frac{\psi_0}{\hbar} \frac{\omega_k - \frac{\hbar}{m}\mathbf{K}\cdot\mathbf{k} + is}{\Omega_k^2 - \left(\frac{\hbar}{m}\mathbf{K}\cdot\mathbf{k} - is\right)^2} \tilde{u}(\mathbf{k})f(s), \quad (18a)$$

$$\tilde{\psi}_1^*(\mathbf{k}, s) = -\frac{\psi_0}{\hbar} \frac{\omega_k + \frac{\hbar}{m}\mathbf{K}\cdot\mathbf{k} - is}{\Omega_k^2 - \left(\frac{\hbar}{m}\mathbf{K}\cdot\mathbf{k} - is\right)^2} \tilde{u}(\mathbf{k})f(s), \quad (18b)$$

where $\hbar\Omega_k = \sqrt{\hbar\omega_k(\hbar\omega_k + 2gn)}$ is the Bogoliubov dispersion. Using the inverse Laplace transformation

$$\frac{\omega_k \mp \frac{\hbar}{m}\mathbf{K}\cdot\mathbf{k} \pm is}{\Omega_k^2 - \left(\frac{\hbar}{m}\mathbf{K}\cdot\mathbf{k} - is\right)^2} \xrightarrow{\mathcal{L}^{-1}} e^{-i\frac{\hbar}{m}\mathbf{K}\cdot\mathbf{k}t}\mathcal{K}^\pm(\mathbf{k}, t), \quad (19)$$

with the abbreviation

$$\mathcal{K}^\pm(\mathbf{k}, t) = \frac{\omega_k}{\Omega_k}\sin(\Omega_k t) \pm i\cos(\Omega_k t), \quad (20)$$

we finally get

$$\tilde{\psi}_1(\mathbf{k}, t) = -\psi_0\tilde{u}(\mathbf{k})A_\mathbf{K}^+(\mathbf{k}, t), \quad (21a)$$

$$\tilde{\psi}_1^*(\mathbf{k}, t) = -\psi_0\tilde{u}(\mathbf{k})A_\mathbf{K}^-(\mathbf{k}, t), \quad (21b)$$

where

$$A_\mathbf{K}^\pm(\mathbf{k}, t) = \int_0^t dt' e^{-i\frac{\hbar}{m}\mathbf{K}\cdot\mathbf{k}t'}\mathcal{K}^\pm(\mathbf{k}, t')f(t - t'), \quad (22)$$

and we used the fact that $\mathcal{K}^\pm(\mathbf{k}, t)$ are functions of $k \equiv |\mathbf{k}|$. Note that $\mathcal{K}^\pm(\mathbf{k}, t)^* = \mathcal{K}^\mp(\mathbf{k}, t)$ and $A_\mathbf{K}^\pm(-\mathbf{k}, t) = A_\mathbf{K}^\mp(\mathbf{k}, t)^* = A_{-\mathbf{K}}^\pm(\mathbf{k}, t)$. Compared to the results of [13], the condensate motion introduces an additional phase factor which depends on $\mathbf{K}\cdot\mathbf{k}$.

In the special case of the condensate at rest, i.e., $\mathbf{K} = \mathbf{0}$, inserting (21) into (8) reproduces the previous result for the condensate deformation [13]

$$q(t) = n\int_{\mathbb{R}^3}\frac{d^3k}{(2\pi)^3}\tilde{\mathcal{R}}(\mathbf{k})|A_\mathbf{0}^-(\mathbf{k}, t)|^2, \quad (23)$$

where we used $A_\mathbf{0}^-(\mathbf{k}, t) = A_\mathbf{0}^+(\mathbf{k}, t)^*$. In addition, the connected correlation function up to the second order becomes

$$\langle\psi(\mathbf{x}, t)\psi^*(\mathbf{y}, t + T)\rangle_c = n\int_{\mathbb{R}^3}\frac{d^3k}{(2\pi)^3}\tilde{\mathcal{R}}(\mathbf{k})e^{i\mathbf{k}\cdot(\mathbf{x}-\mathbf{y})}A_\mathbf{0}^+(\mathbf{k}, t)A_\mathbf{0}^-(\mathbf{k}, t + T). \quad (24)$$

It depends on $\mathbf{x} - \mathbf{y}$ and is therefore translation invariant due to the restored homogeneity. In the general $\mathbf{K} \neq \mathbf{0}$ case, we find the momentum density deformation

$$\langle \delta \mathbf{p}(t) \rangle = n \int_{\mathbb{R}^3} \frac{d^3 k}{(2\pi)^3} \, \tilde{\mathcal{R}}(\mathbf{k}) \hbar \mathbf{k} |A_{\mathbf{K}}^-(\mathbf{k}, t)|^2 \,, \tag{25}$$

as well as the superfluid deformation tensor

$$\delta n_{s,ij}(t) = \frac{n}{\hbar} \int_{\mathbb{R}^3} \frac{d^3 k}{(2\pi)^3} \, \tilde{\mathcal{R}}(\mathbf{k}) \hbar k_i \frac{\partial |A_{\mathbf{K}}^-(\mathbf{k}, t)|^2}{\partial K_j} \bigg|_{\mathbf{K}=\mathbf{0}} \,. \tag{26}$$

Note that the above integrand is proportional to $k_i k_j$ since one has

$$\frac{\partial |A_{\mathbf{K}}^-(\mathbf{k}, t)|^2}{\partial K_j} \bigg|_{\mathbf{K}=\mathbf{0}} = i \frac{\hbar k_j}{m} \int_0^t dt' t' \left[ A_{\mathbf{0}}^-(\mathbf{k}, t) \mathcal{K}^+(\mathbf{k}, t') - A_{\mathbf{0}}^-(\mathbf{k}, t)^* \mathcal{K}^-(\mathbf{k}, t') \right] f(t - t') \,. \tag{27}$$

In this work we are interested in an isotropic system, so the only relevant elements of the superfluid deformation tensor are diagonal and equal, i.e., $\delta n_{s,ij}(t) = \delta n_s(t) \delta_{ij}$. Thus, $\delta n_s(t)$ will be simply referred to as superfluid deformation.

# 3 Time-dependent superfluid deformation

In the following, we will consider the exponential-type disorder ramp-up protocol of [13]

$$f(t) = 1 - e^{-t/\tau} \,, \tag{28}$$

where $\tau$ is the characteristic ramp-up time. For small values of $\tau \ll \tau_{\mathrm{MF}}$, the quenched disorder limit is approached, while for large $\tau \gg \tau_{\mathrm{MF}}$ the equilibrium is reached adiabatically. We will refer to the two aforementioned limits as $\tau \to 0$ and $\tau \to \infty$, respectively. In this exponential-type scenario, (22) becomes

$$A_{\mathbf{K}}^\pm(\mathbf{k}, t) = \frac{\pm \frac{\hbar}{m} \mathbf{K} \cdot \mathbf{k} - \omega_k}{\hbar \left[ \left( \frac{\hbar}{m} \mathbf{K} \cdot \mathbf{k} \right)^2 - \Omega_k^2 \right]} - \frac{i(\omega_k \pm \Omega_k) e^{-it\left( \frac{\hbar}{m} \mathbf{K} \cdot \mathbf{k} + \Omega_k \right)}}{2\hbar \Omega_k \left( \frac{\hbar}{m} \mathbf{K} \cdot \mathbf{k} + \Omega_k \right) \left[ \tau \left( \frac{\hbar}{m} \mathbf{K} \cdot \mathbf{k} + \Omega_k \right) + i \right]}$$

$$+ \frac{\tau e^{-t/\tau} \left[ \tau \omega_k \mp \left( i + \tau \frac{\hbar}{m} \mathbf{K} \cdot \mathbf{k} \right) \right]}{\hbar \left[ \left( i + \tau \frac{\hbar}{m} \mathbf{K} \cdot \mathbf{k} \right)^2 - \tau^2 \Omega_k^2 \right]} + \frac{i(\omega_k \mp \Omega_k) e^{-it\left( \frac{\hbar}{m} \mathbf{K} \cdot \mathbf{k} - \Omega_k \right)}}{2\hbar \Omega_k \left( \frac{\hbar}{m} \mathbf{K} \cdot \mathbf{k} - \Omega_k \right) \left[ \tau \left( \frac{\hbar}{m} \mathbf{K} \cdot \mathbf{k} - \Omega_k \right) + i \right]} \,, \tag{29}$$

which will be used in the sequel. Note that the first term corresponds to the equilibrium value as $\tau \to \infty$, while the rest is the dynamically introduced complement. We will first discuss the case of the arbitrary disorder correlation function $\tilde{\mathcal{R}}(\mathbf{k})$ and then examine the delta-correlated scenario.

## 3.1 Arbitrary correlated disorder

In the case of arbitrary correlated disorder, using (26)–(29) we obtain the full time-dependent superfluid deformation

$$\delta n_{s,\tau}(t) = \frac{n}{m} \int_{\mathbb{R}^3} \frac{d^3 k}{(2\pi)^3} \, \tilde{\mathcal{R}}(\mathbf{k}) k^2 \left[ \frac{4\omega_k}{\Omega_k^4} - \frac{2\tau^4 \omega_k}{\left( \tau^2 \Omega_k^2 + 1 \right)^2} - \frac{4\tau^4 \omega_k e^{-t/\tau}}{\left( \tau^2 \Omega_k^2 + 1 \right)^2} + \frac{2\tau^4 \omega_k e^{-2t/\tau}}{\left( \tau^2 \Omega_k^2 + 1 \right)^2} \right.$$

$$+ \frac{2\omega_k \left( t\tau^3 \Omega_k^4 + \tau(t - 4\tau) \Omega_k^2 - 2 \right) \cos(t\Omega_k)}{\Omega_k^4 \left( \tau^2 \Omega_k^2 + 1 \right)^2} - \frac{2\omega_k \left( \tau^2 (t + 3\tau) \Omega_k^2 + t + \tau \right) \sin(t\Omega_k)}{\Omega_k^3 \left( \tau^2 \Omega_k^2 + 1 \right)^2}$$

$$\left. - \frac{2t\tau \omega_k e^{-t/\tau} \cos(t\Omega_k)}{\Omega_k^2 \left( \tau^2 \Omega_k^2 + 1 \right)} + \frac{2\tau \omega_k e^{-t/\tau} \left( 3\tau^2 \Omega_k^2 + 1 \right) \sin(t\Omega_k)}{\Omega_k^3 \left( \tau^2 \Omega_k^2 + 1 \right)^2} \right] \,. \tag{30}$$

In the long-time limit, the exponentially decaying terms vanish as well as the **k**-space integrals of the oscillatory terms, which leads to the stationary superfluid deformation

$$\delta n_{s,\tau} \equiv \lim_{t\to\infty} \delta n_{s,\tau}(t) = \frac{n}{m} \int_{\mathbb{R}^3} \frac{d^3 k}{(2\pi)^3} \, \tilde{\mathcal{R}}(\mathbf{k}) k^2 \left[ \frac{2\omega_k}{\Omega_k^4} + \frac{2\omega_k\big(2\tau^2\Omega_k^2 + 1\big)}{\Omega_k^4\big(\tau^2\Omega_k^2 + 1\big)^2} \right]. \tag{31}$$

The function in the brackets above is positive and strictly decreasing for $\tau \geq 0$. It starts from the maximum value $4\omega_k/\Omega_k^4$ as $\tau = 0$ and decays towards the asymptotic value $2\omega_k/\Omega_k^4$ for $\tau \to \infty$. The first value corresponds to the quenched regime, where we have

$$\lim_{\tau\to 0} \delta n_{s,\tau} = \frac{n}{m} \int_{\mathbb{R}^3} \frac{d^3 k}{(2\pi)^3} \, \tilde{\mathcal{R}}(\mathbf{k}) \frac{4\hbar^2 k^2}{\omega_k(\hbar\omega_k + 2gn)^2} \,. \tag{32}$$

For adiabatic switch on of the disorder, on the other hand, the asymptotic equilibrium value is reached and it satisfies the general relation

$$\lim_{\tau\to\infty} \delta n_{s,\tau} = \frac{1}{2} \lim_{\tau\to 0} \delta n_{s,\tau} \,. \tag{33}$$

The considered scenario suggests that for an arbitrary disorder correlation $\tilde{\mathcal{R}}(\mathbf{k})$ and a generic dynamical disorder switch-on protocol $f(t)$, the final superfluid deformation has an *equilibrium part* and a *dynamically induced part*. For the exponential ramp-up, the latter part can be at most as large as the equilibrium part. Thus, any excess of the superfluid deformation over the equilibrium value is a hallmark of the non-equilibrium steady state, in the same way as the condensate deformation [13].

## 3.2  Delta-correlated disorder

Let us now specialize in the delta-correlated disorder, which is characterized by

$$\tilde{\mathcal{R}}(\mathbf{k}) = R \,. \tag{34}$$

It can be realized experimentally via a random distribution of many neutral atomic impurities trapped in a deep optical lattice [19–21].

The analytical evaluation of the integral (30) for the superfluid deformation is quite demanding, since the integrand depends both algebraically and trigonometrically on the wave vector. For this purpose, we use a convenient method based on contour integration around a branch cut as described in detail in Appendix A. We assume that temporal quantities are rescaled according to

$$t \to t/\tau_{\mathrm{MF}} \,, \qquad \tau \to \tau/\tau_{\mathrm{MF}} \,, \tag{35}$$

so that $\tau_{\mathrm{MF}}$ will be the unit of time in the rest of the article. In this way, we obtain the full time-dependent expression for the superfluid deformation

$$\frac{\delta n_{s,\tau\neq 1}(t)}{q_{\mathrm{HM}}} = \frac{8\sqrt{t}e^{-t}}{3\sqrt{\pi}\,(\tau^2 - 1)} \Big[\tau\big(e^{-t/\tau} - 1\big) - 1\Big] - \frac{2(2\tau + 3)\sqrt{\tau}}{3(\tau + 1)^{3/2}} \, \mathrm{erf}\left(\sqrt{t/\tau}\sqrt{\tau + 1}\right)$$

$$+ \frac{8}{3}\,\mathrm{erf}\left(\sqrt{t}\right) - \frac{2(2\tau - 3)\sqrt{\tau}e^{-t/\tau}}{3(\tau - 1)^{3/2}}\big(2 - e^{-t/\tau}\big)\,\mathrm{erf}\left(\sqrt{t/\tau}\sqrt{\tau - 1}\right), \tag{36}$$

which is valid for $\tau \neq 1$ and $\mathrm{erf}(x)$ denotes the error function. For $\tau = 1$ we obtain

$$\frac{\delta n_{s,\tau=1}(t)}{q_{\mathrm{HM}}} = \frac{16\,\mathrm{erf}\left(\sqrt{t}\right) - 5\sqrt{2}\,\mathrm{erf}\left(\sqrt{2t}\right)}{6} + \frac{16e^{-t}(t - 3)\sqrt{t}}{9\sqrt{\pi}} + \frac{2e^{-2t}\sqrt{t}(15 - 4t)}{9\sqrt{\pi}} \,, \tag{37}$$

in units of the Huang-Meng equilibrium condensate deformation [14]

$$q_{\mathrm{HM}} = R \frac{m^{3/2}\sqrt{n}}{4\pi\hbar^3\sqrt{g}}. \tag{38}$$

This result is graphically represented in Fig. 1(a). Initially, at $t = 0$, the system is a condensate, so the superfluid deformation is zero. Based on (36) and (37), the superfluid deformation at early times $t \to 0$ behaves asymptotically as

$$\frac{\delta n_{s,\tau}(t)}{q_{\mathrm{HM}}} = \frac{16t^{5/2}}{5\sqrt{\pi}\tau^2} - \frac{8t^{7/2}(10\tau + 21)}{63\sqrt{\pi}\tau^3} + \dots, \tag{39}$$

which is dominated by a $t^{5/2}$ power-law. Conversely, in the long-time limit $t \to \infty$ also the stationary superfluid deformation is accessible from (36) and (37) and reads

$$\frac{\delta n_{s,\tau}}{q_{\mathrm{HM}}} = \frac{8}{3} - \frac{2\sqrt{\tau}(2\tau + 3)}{3(\tau + 1)^{3/2}}, \tag{40}$$

which is depicted in Fig. 1(b). In the sudden-quench scenario, we get

$$\lim_{\tau \to 0} \delta n_{s,\tau} = \frac{8}{3}q_{\mathrm{HM}}, \tag{41}$$

while for the adiabatic switch on of the disorder we find

$$\lim_{\tau \to \infty} \delta n_{s,\tau} = \frac{4}{3}q_{\mathrm{HM}}, \tag{42}$$

which is precisely the equilibrium Huang-Meng result [14].

Let us now investigate how the superfluid deformation relaxes to the long-time limit $t \to \infty$. From (36) and (37) we get

$$\frac{\delta n_{s,\tau}(t) - \delta n_{s,\tau}}{q_{\mathrm{HM}}} \approx \begin{cases} B_I(t,\tau)e^{-t}, & \text{if } 0 < \tau < 1, \\ B_{II}(t,\tau)e^t, & \text{if } \tau = 1, \\ B_{III}(\tau)e^{-t/\tau}, & \text{if } \tau > 1, \end{cases} \tag{43}$$

where the respective prefactors are given explicitly as follows:

$$B_I(t,\tau) = \frac{4[2t(1-\tau) + \tau - 2]}{3\sqrt{\pi}\sqrt{t}(1-\tau)^2}, \tag{44a}$$

$$B_{II}(t,\tau) = \frac{8[2t(t-3) - 3]}{9\sqrt{\pi}\sqrt{t}}, \tag{44b}$$

$$B_{III}(\tau) = \frac{4(3 - 2\tau)\sqrt{\tau}}{3(\tau - 1)^{3/2}}. \tag{44c}$$

We observe that the superfluid deformation approaches the asymptotic regime exponentially on a unit time scale for $0 < \tau \leq 1$ and otherwise on a time scale of $\tau$. Recalling that the unit of time is $\tau_{\mathrm{MF}}$, we can conclude that the asymptotic behavior is determined by the larger of the two timescales involved: the mean-field-related or the switch-on-protocol-induced. Further elaboration on this point can be found at the end of Appendix A. We note that (43) and (44) are valid for any *fixed value* of $\tau$. The apparent divergence of $B_I(t,\tau)$ and $B_{III}(t,\tau)$ as $\tau \to 1$ arises because this limit conflicts with the assumption that the argument of the final error function in (36) is asymptotically large.

# 4 Time-dependent condensate deformation

In this section, we examine in the same way the full-time dependence of the condensate deformation. In the above scenario, from (23), (29), and (34) follows

$$
q_\tau(t) = \frac{nR}{\hbar^2} \int_{\mathbb{R}^3} \frac{d^3k}{(2\pi)^3} \left\{ \frac{\omega_k^2}{\Omega_k^4} + \frac{\omega_k^2 + \Omega_k^2}{2\Omega_k^4 \left(1+\tau^2\Omega_k^2\right)} - \frac{2\omega_k^2 \cos(\Omega_k t)}{\Omega_k^4 \left(1+\tau^2\Omega_k^2\right)} - \frac{2\tau\omega_k^2 \sin(\Omega_k t)}{\Omega_k^3 \left(1+\tau^2\Omega_k^2\right)} \right.
$$
$$
+ \frac{\left(\Omega_k^2 - \omega_k^2\right)\left(\tau^2\Omega_k^2 - 1\right)\cos(2\Omega_k t)}{2\Omega_k^4 \left(1+\tau^2\Omega_k^2\right)^2} - \frac{\tau\left(\Omega_k^2 - \omega_k^2\right)\sin(2\Omega_k t)}{\Omega_k^3 \left(1+\tau^2\Omega_k^2\right)^2} + e^{-2t/\tau}\frac{\tau^2\left(1+\tau^2\omega_k^2\right)}{\left(1+\tau^2\Omega_k^2\right)^2}
$$
$$
+ e^{-t/\tau}\left[\frac{2\tau^2\left(\omega_k^2 - \Omega_k^2\right)\cos(\Omega_k t)}{\Omega_k^2 \left(1+\tau^2\Omega_k^2\right)^2} - \frac{2\tau^2\omega_k^2}{\Omega_k^2 \left(1+\tau^2\Omega_k^2\right)} + \frac{2\tau\left(1+\tau^2\omega_k^2\right)\sin(\Omega_k t)}{\Omega_k \left(1+\tau^2\Omega_k^2\right)^2}\right]\right\}, \quad (45)
$$

which coincides with Eq. (21) of Ref. [13]. Previously, the above integral was only determined numerically, but now we can solve it analytically using the method of contour integration around a branch cut from Appendix A. We obtain the closed-form expression

$$
\frac{q_\tau(t)}{q_{\mathrm{HM}}} = \frac{4\sqrt{t}\tau(2\tau - 1)e^{-t(\tau+1)/\tau}}{\sqrt{\pi}(\tau - 1)} + \frac{4e^{-t}\sqrt{t}(1 - 2t + 4\tau)}{\sqrt{\pi}}
$$
$$
+ 2\left[1 - 4t^2 - 8\tau^2\left(e^{-t/\tau} + 1\right) + 8\tau t\right]\mathrm{erf}\left(\sqrt{t}\right) + \left(8t^2 + 12\tau^2 - 16\tau t + \frac{1}{2}\right)\mathrm{erf}\left(\sqrt{2t}\right)
$$
$$
+ \frac{2\sqrt{\tau}e^{-t/\tau}}{\sqrt{\tau - 1}}\left[(2\tau - 1)\left(e^{-t/\tau}(2t + 3\tau) + 4\tau\right) - 2 - \frac{e^{-t/\tau}}{\tau - 1}\right]\mathrm{erf}\left(\frac{\sqrt{t(\tau - 1)}}{\sqrt{\tau}}\right)
$$
$$
- \frac{\sqrt{\tau}e^{-2t/\tau}}{(\tau - 1)^{3/2}}\left[4t(\tau - 1)(2\tau - 1) + \tau(6\tau(2\tau - 3) + 5)\right]\mathrm{erf}\left(\frac{\sqrt{2t(\tau - 1)}}{\sqrt{\tau}}\right)
$$
$$
+ \sqrt{\tau(\tau + 1)}(4\tau - 2)\,\mathrm{erf}\left(\frac{\sqrt{t(\tau + 1)}}{\sqrt{\tau}}\right) + \frac{\sqrt{2t}e^{-2t}}{\sqrt{\pi}(\tau - 1)}\left[4t(\tau - 1) + 3(3 - 4\tau)\tau + 1\right], \quad (46)
$$

which is also plotted in Fig. 1(a). This analytic result enables us to reveal the condensate deformation behavior in several regimes and compare it with the superfluid deformation (36) and (37). For example, for small times $t \to 0$ we find that the condensate deformation has the same $t^{5/2}$-dominated power-law behavior

$$
\frac{q_\tau(t)}{q_{\mathrm{HM}}} = \frac{16t^{5/2}}{5\sqrt{\pi}\tau^2} + \frac{8\left[\left(32\sqrt{2} - 66\right)\tau - 35\right]t^{7/2}}{105\sqrt{\pi}\tau^3} + \dots, \quad (47)
$$

as the superfluid deformation (39). In the long-time limit we recover the previous result [13, Eqs. (27) and (28)]

$$
q_\tau = \lim_{t\to\infty} q_\tau(t) = \left[\frac{5}{2} - 4\tau^2 + 2\left(2\tau - 1\right)\sqrt{\tau(\tau + 1)}\right]q_{\mathrm{HM}}, \quad (48)
$$

and obtain that the condensate deformation approaches the asymptotic limit $t \to \infty$ as

$$
\frac{q_\tau(t) - q_\tau}{q_{\mathrm{HM}}} \approx \begin{cases} C_I(t, \tau)e^{-t}, & \text{if } 0 < \tau < 1, \\ C_{II}(t)e^{-t}, & \text{if } \tau = 1, \\ C_{III}(\tau)e^{-t/\tau}, & \text{if } \tau > 1. \end{cases} \quad (49)
$$

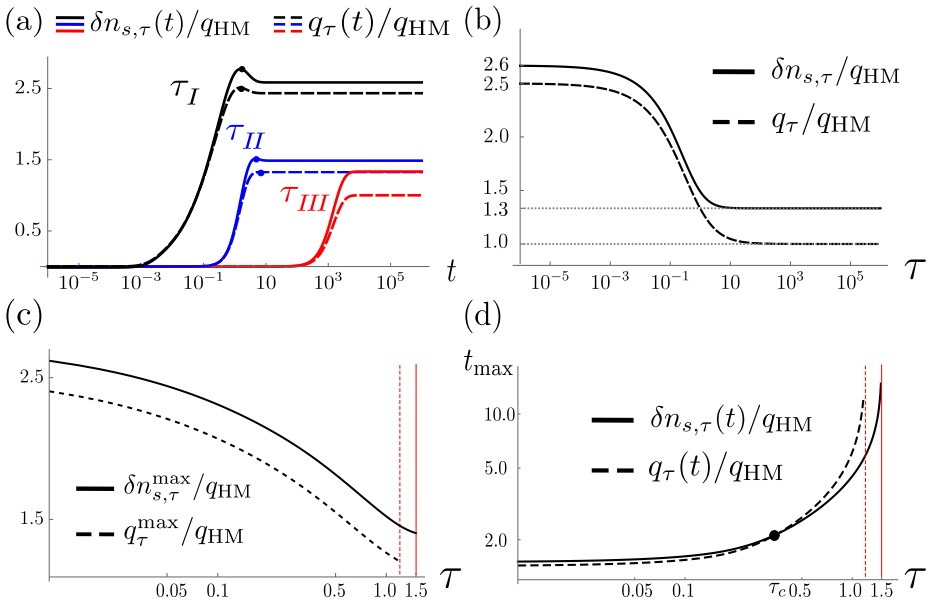

Figure 1: (a) The superfluid deformation (solid lines) and condensate deformation (dashed lines) as a function of the rescaled time $t$ for three values of the rescaled disorder ramp-up time $\tau_I = 10^{-3}$, $\tau_{II} = 1$ and $\tau_{III} = 10^3$. Both quantities are normalized by the equilibrium Huang-Meng condensate deformation for the delta-correlated disorder ($q_{\mathrm{HM}}$, see main text). (b) The stationary superfluid deformation (solid line) and stationary condensate deformation (dashed line) as a function of $\tau$. (c) The maximum, and (d) the time corresponding to the maximum of the superfluid (solid lines) and condensate (dashed lines) deformations as function of $\tau$, exemplified by the dots in (a). The vertical lines indicate the maximal ramp-up times after which the overshooting above the stationary value does not occur. The two curves in (d) intersect at $\tau_c \approx 0.35$.

The three corresponding amplitudes are given by

$$C_I(t,\tau) = -\frac{4}{\sqrt{\pi}\sqrt{t}(\tau-1)}\,, \tag{50a}$$

$$C_{II}(t) = -\frac{34}{\sqrt{\pi}t^{3/2}} + \frac{8\sqrt{t}}{\sqrt{\pi}} + \frac{28}{\sqrt{\pi}\sqrt{t}} - 16\,, \tag{50b}$$

$$C_{III}(\tau) = \frac{4\sqrt{\tau}\,[2\tau(2\tau-1)-1]}{\sqrt{\tau-1}} - 16\tau^2\,. \tag{50c}$$

We notice that the superfluid deformation is slightly more sensitive than the condensate deformation to the presence of the disorder, see Fig. 1(a)-(b). However, both superfluid and condensate deformations approach their respective equilibrium on the same time scale, showing different algebraic amplitude dependencies. As in the case of superfluid deformation, the asymptotic behavior of condensate deformation is determined by the larger of the two timescales: the mean-field-related or the switch-on-protocol-induced. The next section will discuss the relationship between condensate deformation and a specific disorder-averaged correlation function.

Before moving on, we would like to point out a particular dynamical feature of both superfluid and condensate deformations, as seen in Fig. 1(a). Specifically, for ramp-up times $\tau \lesssim 1.5$, both quantities overshoot their stationary values. They reach a maximum at a certain point, as indicated by the dots, and then relax to the steady state. We explore

such behavior in detail in Fig. 1(c)-(d). The former panel displays how the maximum values depend on $\tau$, while the latter shows when the respective maximums appear, i.e., the maximal overshooting times $t_{\max}$. For both quantities, there exists a certain maximal value of $\tau$, after which the overshooting no longer occurs. As indicated by the vertical lines, the aforementioned overshooting first ends for the condensate deformation around $\tau \approx 1.2$. For the superfluid deformation, it ends around $\tau \approx 1.5$. Panel (d) shows that for $\tau < \tau_{\mathrm{c}} \approx 0.35$, $q_\tau(t)$ reaches its maximum slightly before $\delta n_{s,\tau}(t)$, while the opposite is true for $\tau > \tau_{\mathrm{c}}$. The maximums appear within the intermediate time window $1 \lesssim t_{\max} \lesssim 10$. One plausible physical mechanism behind the overshooting is the excitation of high-energy Bogoliubov modes when the ramp-up protocol is rapid enough. We verified that overshooting is still present in the linear ramp-up protocol. Consequently, this qualitative feature appears to be protocol-independent. A more detailed understanding of the underlying mechanisms would provide valuable insight into the corresponding intermediate dynamics, which is left as future work.

# 5 Stationary correlation functions

We now analyze the stationary connected correlation function $\lim_{t\to\infty}\langle\psi(\mathbf{x},t)\psi^*(\mathbf{y},t+T)\rangle_c$ (13), which could be accessed experimentally. To this end, the atomic gas is split into two clouds, which are then allowed to expand. Then the interference probe is used with a matter wave heterodyning and the coherence of the system is encoded in the first order correlation function [22]. Focusing on the long-time limit regime of the connected correlation function (24), the time integrals (22) have to be evaluated. Taking into account the disorder ramp-up protocol (28), regardless of the random potential, the following result is obtained up to the second order

$$\lim_{t\to\infty} A_{\mathbf{0}}^+(\mathbf{k},t)A_{\mathbf{0}}^-(\mathbf{k},t+T) = \frac{\omega_k^2}{\hbar^2\Omega_k^4} + \frac{e^{iT\Omega_k}(\omega_k+\Omega_k)^2}{4\hbar^2\Omega_k^4\left(\tau^2\Omega_k^2+1\right)} + \frac{e^{-iT\Omega_k}(\omega_k-\Omega_k)^2}{4\hbar^2\Omega_k^4\left(\tau^2\Omega_k^2+1\right)}. \quad (51)$$

In the above equation, the exponentially decaying terms have been safely neglected, and the remaining terms involving trigonometric functions turn out to have no contribution in the stationary regime.

In the following, we consider the case of delta-correlated disorder (34). The corresponding stationary expression for the connected correlation (24) is then

$$\langle\psi(\mathbf{z})\psi^*(T)\rangle_c = nR\int_{\mathbb{R}^3}\frac{d^3k}{(2\pi)^3}e^{i\mathbf{k}\cdot\mathbf{z}}\left[\frac{\omega_k^2}{\hbar^2\Omega_k^4} + \frac{e^{iT\Omega_k}(\omega_k+\Omega_k)^2}{4\hbar^2\Omega_k^4\left(\tau^2\Omega_k^2+1\right)} + \frac{e^{-iT\Omega_k}(\omega_k-\Omega_k)^2}{4\hbar^2\Omega_k^4\left(\tau^2\Omega_k^2+1\right)}\right], \quad (52)$$

where we use the shorthand notation $\langle\psi(\mathbf{z}=\mathbf{x}-\mathbf{y})\psi^*(T)\rangle_c = \lim_{t\to\infty}\langle\psi(\mathbf{x},t)\psi^*(\mathbf{y},t+T)\rangle_c$ hereafter. Note that the above connected correlation consists of an equilibrium and a dynamically induced part in the same manner as other quantities of interest. Moreover, for the delta-correlated disorder, it really only depends on $|\mathbf{z}|$. Note that we cannot evaluate the integral in (52) for arbitrary $\mathbf{z}$ and $T$. Therefore, in the following subsections we examine two cases that are analytically accessible. On the one hand, we consider the equal-time connected correlation function, i.e., we set $T = 0$, and on the other hand, we discuss the equal-space connected correlation function, i.e., we set $\mathbf{z} = \mathbf{0}$.

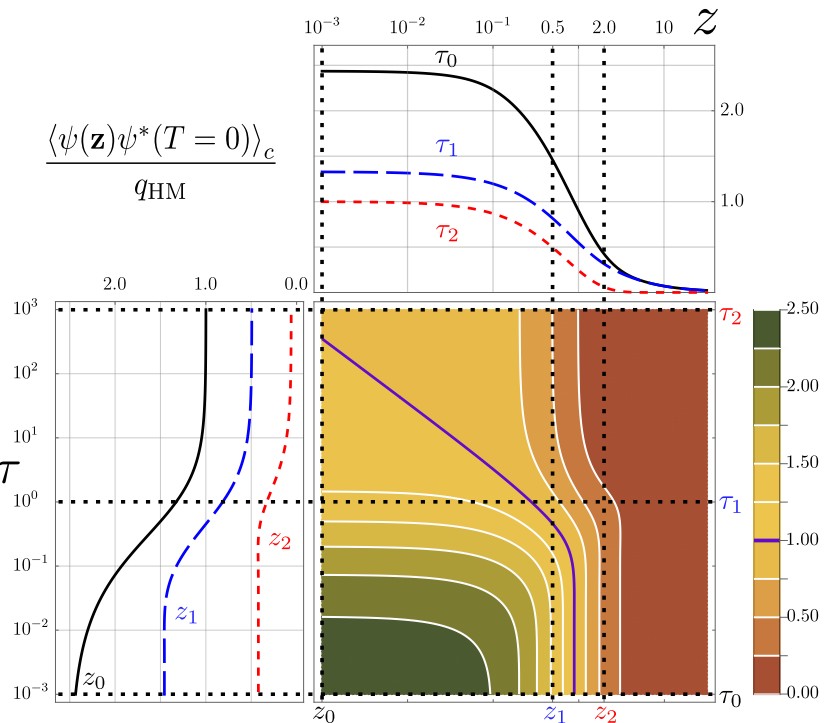

Figure 2: Density plot representing the stationary equal-time two-point correlation function (54) normalized by the equilibrium condensate deformation $q_{\mathrm{HM}}$ in terms of rescaled spatial separation $z$ and the rescaled characteristic ramp-up time $\tau$, both in logarithmic scale. Equal-time two-point correlation (left panel) as a function of $\tau$ for three values $10^3 z_0 = 2z_1 = z_2/2 = 1$ and (top panel) as a function of $z$ for three values $10^3 \tau_0 = \tau_1 = 10^{-3} \tau_2 = 1$. The purple curve corresponds to $\langle \psi(\mathbf{z})\psi^*(T=0)\rangle_c = q_{\mathrm{HM}}$.

## 5.1 Equal-time two-point connected correlation function

The $T = 0$ equal-time connected correlation (52) is real and after angular integration reads

$$\langle \psi(\mathbf{z})\psi^*(T=0)\rangle_c = \frac{2nR}{(2\pi)^2} \int_0^\infty dk\, k \frac{\sin(k|\mathbf{z}|)}{|\mathbf{z}|}\left[ \frac{\omega_k^2}{\hbar^2\Omega_k^4} + \frac{\omega_k^2 + \Omega_k^2}{2\hbar^2\Omega_k^4\left(\Omega_k^2\tau^2 + 1\right)}\right]. \tag{53}$$

Evaluation of the above integral yields

$$\langle \psi(\mathbf{z})\psi^*(T=0)\rangle_c = q_{\mathrm{HM}}\left\{ \frac{3}{2}e^{-\sqrt{2}z} + \frac{1 - e^{-\sqrt{2}z}}{\sqrt{2}z} + \frac{2\sqrt{2}\tau^2 e^{-\sqrt{2}z}}{z} + \frac{2\sqrt{2}}{z}e^{-\frac{z\sqrt{\tau+1}}{\sqrt{2\tau}}} \right.$$
$$\left. \times \left[ \tau\sqrt{\tau^2-1}\sinh\left(\frac{z\sqrt{\tau-1}}{\sqrt{2\tau}}\right) - \tau^2\cosh\left(\frac{z\sqrt{\tau-1}}{\sqrt{2\tau}}\right)\right]\right\}, \tag{54}$$

where we introduced the dimensionless spatial separation $z = |\mathbf{z}|/\xi$ rescaled by the healing length $\xi = \hbar/\sqrt{2gnm}$.

First, we plot the equal-time two-point correlation function in the top panel of Fig. 2. We observe that for a fixed ramp-up time $\tau$ it decreases monotonically in space and interpolates between the sudden quench and an adiabatic ramp-up of the disorder, respectively,

$$\lim_{\tau\to 0}\langle \psi(\mathbf{z})\psi^*(T=0)\rangle_c = q_{\mathrm{HM}}\left[ \frac{3}{2}e^{-\sqrt{2}z} + \frac{1 - e^{-\sqrt{2}z}}{\sqrt{2}z}\right], \tag{55}$$

$$\lim_{\tau\to\infty}\langle \psi(\mathbf{z})\psi^*(T=0)\rangle_c = q_{\mathrm{HM}}\,e^{-\sqrt{2}z}. \tag{56}$$

We notice that in the former case the correlation asymptotically decays to zero algebraically as $\propto 1/z$, while in the latter case the decay is exponential. This shows that the quench produces long-range spatial correlations, as opposed to rapidly decaying equilibrium correlations.

Furthermore, the left panel of Fig. 2 reveals that for any fixed distance $z$ the connected correlation function decreases monotonically with the ramp-up time $\tau$. On the one hand, there is no off-diagonal long-range order

$$\lim_{z \to \infty} \langle \psi(\mathbf{z})\psi^*(T=0)\rangle_c = 0, \tag{57}$$

i.e., the long-range spatial stationary correlations of the disordered Bose gas vanish. On the other hand, when the two points spatially coincide, we find

$$\lim_{z \to 0} \langle \psi(\mathbf{z})\psi^*(T=0)\rangle_c = q_\tau. \tag{58}$$

In this way, we obtain a connection between the equal-time two-point correlation function and the stationary condensate deformation (48) for any ramp-up time $\tau$. This analysis is consistent with the generic drop of (54) along the diagonal of the central panel of Fig. 2.

## 5.2    Equal-space connected correlation function

The $\mathbf{k}$-integral in the equal-space connected correlation (52) can be solved analytically for $\mathbf{z} = \mathbf{0}$ using the method developed in Appendix A. The result reads

$$
\begin{aligned}
\frac{\langle \psi(\mathbf{z}=\mathbf{0})\psi^*(T)\rangle_c}{q_{\text{HM}}} &= 1 + \frac{\sqrt{T}e^{-T}}{\sqrt{\pi}}(2T - 1 + 4i) - \left[4\tau^2 + 2T(T+2i) - \frac{3}{2}\right]\text{erfc}\left(\sqrt{T}\right) \\
&+ \frac{\sqrt{2\tau}\left(\sqrt{\tau^2-1}+i\right)e^{-T/\tau}}{\sqrt{\tau^2-1}\left(\sqrt{\tau^2-1}+\tau\right)^{3/2}} + \frac{\sqrt{\tau}e^{-T/\tau}}{\sqrt{\tau-1}}\left[2\tau^2 - (1+2i)\tau - 1 + i\right]\text{erfc}\left(\frac{\sqrt{T(\tau-1)}}{\sqrt{\tau}}\right) \\
&+ \frac{\sqrt{\tau}e^{T/\tau}}{\sqrt{\tau+1}}\left[2\tau^2 + (1+2i)\tau - 1 + i\right]\text{erfc}\left(\frac{\sqrt{T(\tau+1)}}{\sqrt{\tau}}\right),
\end{aligned}
\tag{59}
$$

where $\text{erfc}(x) = 1 - \text{erf}(x)$ is the complementary error function and similarly as in (35) the rescaling of the time delay $T \to T/\tau_{\text{MF}}$ is being employed. The behavior of the equal-space connected correlation (59) is presented in Fig. 3. In part (a) we plot its amplitude normalized by the equilibrium condensate deformation $q_{\text{HM}}$, while in part (b) we display its phase. The top panels of parts (a) and (b) reflect the fact that the amplitude decreases monotonically from the equal-time value

$$\langle \psi(\mathbf{z}=\mathbf{0})\psi^*(T=0)\rangle_c = q_\tau, \tag{60}$$

which is precisely the stationary condensate deformation, towards the Huang-Meng equilibrium condensate deformation at long time delays

$$\lim_{T \to \infty} \langle \psi(\mathbf{z}=\mathbf{0})\psi^*(T)\rangle_c = q_{\text{HM}}. \tag{61}$$

At the same time, the phase becomes zero in the above two limits, i.e., the equal-space correlation (59) becomes real-valued. At intermediate time delays the connected correlation function is complex and its phase features a maximum.

On the other hand, the left panels of parts (a) and (b) display the amplitude and phase of (59) as functions of ramp-up time for three time delays. The plots manifest the complex-valued sudden quench limit

$$\frac{\lim_{\tau \to 0}\langle \psi(\mathbf{z}=\mathbf{0})\psi^*(T)\rangle_c}{q_{\text{HM}}} = 1 + \frac{\sqrt{T}e^{-T}}{\sqrt{\pi}}(2T - 1 + 4i) - \left[2T(T+2i) - \frac{3}{2}\right]\text{erfc}\left(\sqrt{T}\right), \tag{62}$$

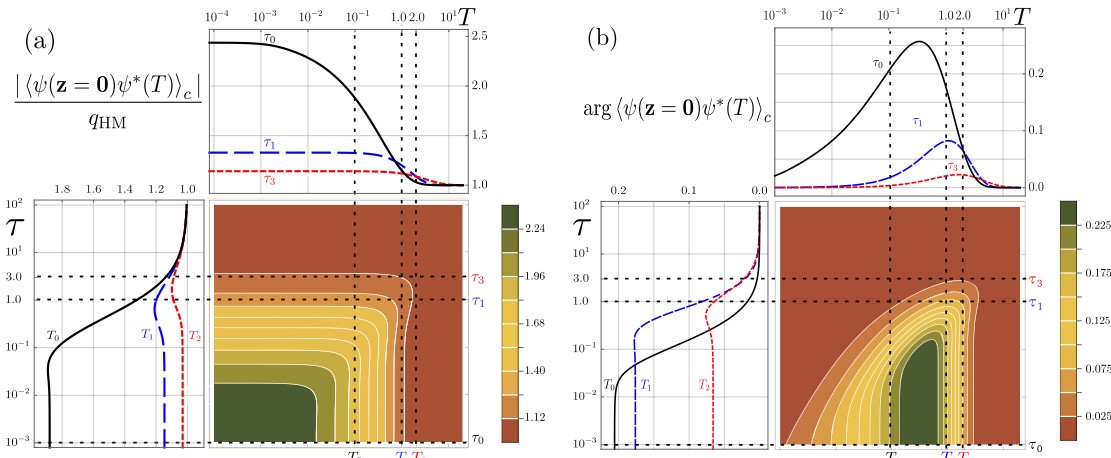

Figure 3: Density plot of the amplitude normalized by the equilibrium condensate deformation $q_{\mathrm{HM}}$ (a) and phase (b) of the equal-space connected correlation (59) as functions of rescaled time delay $T$ and the rescaled ramp-up time $\tau$, both in logarithmic scale. In the left panels we used $10T_0 = T_1 = T_2/2 = 1$, while in the top panels we have $10^3\tau_0 = \tau_1 = \tau_3/3 = 1$.

as well as the equilibrium Huang-Meng limit

$$\lim_{\tau \to \infty} \langle \psi(\mathbf{z} = \mathbf{0})\psi^*(T)\rangle_c = q_{\mathrm{HM}}. \tag{63}$$

In particular, the equal-space equal-time sudden quench yields the maximal stationary condensate deformation

$$\lim_{\tau \to 0} \langle \psi(\mathbf{z} = \mathbf{0})\psi^*(T = 0)\rangle_c = \frac{5}{2}q_{\mathrm{HM}}. \tag{64}$$

Both the amplitude and phase feature a single maximum along any horizontal or vertical cross-section, as exemplified in the central panels of the parts (a) and (b) in Fig. 3. For a sufficiently large ramp-up time $\tau$ and/or time delay $T$, the amplitude approaches the value $q_{\mathrm{HM}}$, while the phase vanishes. The former condition is directly related to the adiabaticity of the driving protocol.

## 6   Summary and outline

In this paper, we continue the study of Ref. [13] on the out-of-equilibrium dynamical properties of Bose-Einstein condensates in a ramped-up weak delta-correlated disorder. On the one hand, we advanced our understanding of the effects of weak disorder by analyzing and discussing more physical observables. On the other hand, we developed a new computational technique to obtain the respective results analytically. In Ref. [13] we found that the emerging stationary condensate deformation, after the onset of the disorder, turns out to be a sum of a reversible equilibrium part, which actually corresponds to the adiabatic switch-on of the disorder, and an irreversible dynamically induced part, which depends on the details of the switch-on protocol. Here we show that a similar decomposition into a reversible and an irreversible contribution occurs for both the correlation function and the superfluid deformation. In the limiting case of equal time and space, the connected correlation functions turn out to coincide with the stationary condensate deformation for any ramp-up time.

These results have direct implications for the scenario examined in Ref. [23], where the disorder is first switched on and then off. The total stationary superfluid deformation $\delta n_{s,\tau_1,\tau_2}$ depends on both the respective time scales $\tau_1$ and $\tau_2$. If both processes occur adiabatically, the final superfluid deformation $\delta n_{s,\infty,\infty}$ vanishes. If one of the two processes is adiabatic and the other is quenched, we get $\delta n_{s,0,\infty} = \delta n_{s,\infty,0} = 4q_{\mathrm{HM}}/3$. And finally, if both the switching on and off of the disorder are implemented as sudden processes, the two irreversible contributions add up so that the total superfluid deformation is $\delta n_{s,0,0} = 8q_{\mathrm{HM}}/3$. Similar conclusions can be drawn for the considered connected correlation functions.

As an outlook, we mention that our results suggest that the disorder ensemble averaged correlation functions, which are now experimentally accessible by new quantum gas microscope detection schemes [24–26], should be explored in more detail. These advances nourish the prospect of also investigating the fluctuation-dissipation theorem for dirty bosons. In this sense, Ref. [27] recently analyzed the Maxwell relation between entropy and atom-atom pair correlation and it serves as a proof of principle demonstration for further experimental developments.

## Acknowledgements

R.P.A.L. is grateful for the hospitality of Technical University of Kaiserslautern and Universidade Federal de Alagoas, where part of this work was carried out.

**Funding information**    This work was supported by the Improvement Coordination of Higher Level Personnel (CAPES) under PROBRAL program Nº 12/2017, CNPq (Conselho Nacional de Desenvolvimento Científico e Tecnológico), and DAAD-CAPES PRO-BRAL, Brazil Grant No. 88887.627948/2021-00. A.P. and M.R. acknowledge financial support by the Deutsche Forschungsgemeinschaft (DFG, German Research Foundation) via the Collaborative Research Center SFB/TR185 (Project No. 277625399) and via the Research Unit FOR 2247 (Project No. PE 530/6-1). M.R. also acknowledges financial support by the DFG via the Research Grant No. 274978739.

## A    Contour integration method

Here we present the method of solving the **k**-integrals when the integrand involves the Bogoliubov dispersion $\Omega_k$ in a more complicated way than the algebraic one, which could be solved straightforwardly by the residue theorem. An example is when the integrand contains a trigonometric function of $\Omega_k$. We encountered such integrals in the time dependence of all quantities of interest. We will present a pedagogical example that includes all the major difficulties we faced in solving such **k**-integrals.

To illustrate our method, we choose Eq. (59) as a pedagogical example. The equal-space connected correlation function is a good illustration with the minimum ingredients necessary to demonstrate our method. In the long-time limit, the equal-space connected correlation function has two terms. The first corresponds to the Huang-Meng contribution associated with the adiabatic switch on of the disorder, while the second is dynamically induced with a trigonometric function depending on $\Omega_k$

$$I_\tau(T) = n \int_{\mathbb{R}^3} \frac{d^3k}{(2\pi)^3} \tilde{\mathcal{R}}(\mathbf{k}) \left[ \frac{\omega_k^2}{\hbar^2 \Omega_k^4} + \frac{e^{-iT\Omega_k}(\omega_k - \Omega_k)^2 + e^{iT\Omega_k}(\omega_k + \Omega_k)^2}{4\hbar^2 \Omega_k^4 \left( \tau^2 \Omega_k^2 + 1 \right)} \right], \qquad (65)$$

which can be cast into a dimensionless form using the replacements $k \to k/\xi$, $T \to T\hbar/gn$, and $\tau \to \tau\hbar/gn$, yielding

$$\frac{I_\tau(T)}{q_{\mathrm{HM}}} = 1 + \int_0^\infty dk \frac{4\sqrt{2}}{\pi} \frac{(k^2+1)\cos\left(Tk\sqrt{k^2+2}\right) + ik\sqrt{k^2+2}\sin\left(Tk\sqrt{k^2+2}\right)}{(k^2+2)^2\left[\tau^2 k^2(k^2+2) + 1\right]}. \tag{66}$$

To solve the above integral, we employ the transformation $x = k\sqrt{k^2+2}$ and get

$$\frac{I_\tau(T)}{q_{\mathrm{HM}}} = 1 + \int_{-\infty}^\infty dx \frac{\sqrt{2}}{\pi} \frac{\left(\sqrt{x^2+1}+x\right)e^{iTx}}{\sqrt{x^2+1}\left(\sqrt{x^2+1}+1\right)^{3/2}(\tau^2 x^2 + 1)}, \tag{67}$$

where we used $\sqrt{x^2+1} - 1 = x^2/(\sqrt{x^2+1}+1)$ and Euler's formula. The lower limit of integration has been extended to $-\infty$, taking into account the parity of different parts of the integrand.

To solve the integral in (67), we apply the Cauchy residue theorem

$$\oint_C h(z)dz = 2\pi i \sum_k \operatorname*{Res}_{z=z_k} h(z), \tag{68}$$

to the complex plane contour $C$ shown in Fig. 4, where

$$h(z) = \frac{\sqrt{2}}{\pi} \frac{\left(\sqrt{z^2+1}+z\right)e^{iTz}}{\sqrt{z^2+1}\left(\sqrt{z^2+1}+1\right)^{3/2}(\tau^2 z^2 + 1)}. \tag{69}$$

We are faced with an algebraic branch cut $[i, +i\infty)$ along the imaginary line due to the aforementioned transformation, which also appears in other methods [28]. The integrand has a simple pole $\gamma$ at $z = i/\tau$ and there are no poles within the contour $C$. For $\tau > 1$ the pole $\gamma$ is outside the branch cut, while for $0 < \tau < 1$ it lies on the cut. In the latter case, one should shift the pole infinitesimally to the right by $\epsilon > 0$ and take the limit $\epsilon \to 0$ in the final step of the calculation.

Let us introduce the real interval $I = (-R, R)$, so that

$$\int_I h(z)dz + \int_{C_R} h(z)dz + \int_{c_r} h(z)dz + \int_{c_\gamma} h(z)dz + \int_{\Gamma_>} h(z)dz + \int_{\Gamma_<} h(z)dz = 0. \tag{70}$$

The integral along the two disconnected parts of the semicircle $C_R$ of radius $R$ tends to zero as $R \to \infty$. The contribution along the circle $c_r$ of radius $r$ around the terminus of

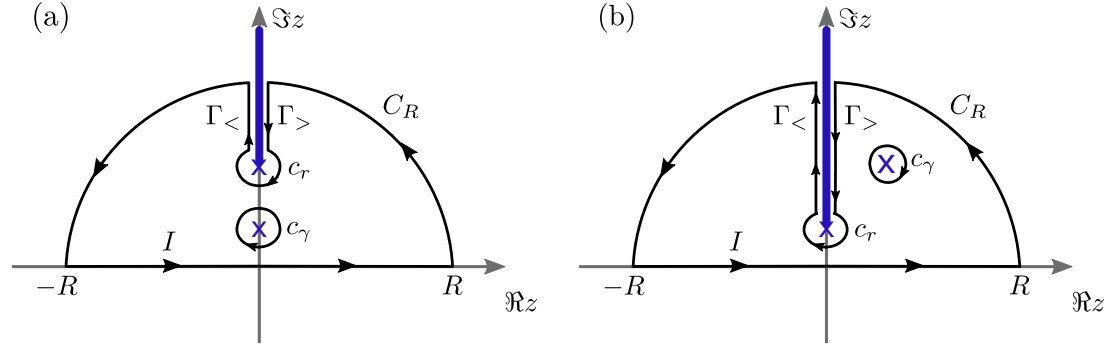

Figure 4: (a) Integration contour $C$ in the complex $z$-plane for $\tau > 1$. (b) For $0 < \tau < 1$ the pole $\gamma$ at $z = i/\tau$ is shifted infinitesimally to the right by $\epsilon > 0$ in order to not lie on the branch cut.

the branch cut at $z = i$ vanishes for $r \to 0$. The integral along the circle $c_\gamma$ is the *negative* residue at $z = i/\tau$. The remaining two integrals along $\Gamma_>$ and $\Gamma_<$ take into account the contribution of the algebraic branch cut. Thus, assuming $R \to \infty$, we arrive at

$$\int_{-\infty}^{\infty} h(z)dz = 2\pi i \operatorname*{Res}_{z=i/\tau} h(z) - \int_{\Gamma_>} h(z)dz - \int_{\Gamma_<} h(z)dz. \tag{71}$$

The residue contribution is

$$2\pi i \operatorname*{Res}_{z=i/\tau} h(z) = \frac{\sqrt{2\tau}\left(\sqrt{\tau^2 - 1} + i\right)e^{-T/\tau}}{\sqrt{\tau^2 - 1}\left(\sqrt{\tau^2 - 1} + \tau\right)^{3/2}}, \tag{72}$$

while the integration around the branch cut yields

$$
\begin{aligned}
\int_{\Gamma_>} h(z)dz + \int_{\Gamma_<} h(z)dz &= \lim_{\varepsilon \to 0^+} i\left[\int_{\infty}^{1} h(i\eta + \varepsilon)d\eta + \int_{1}^{\infty} h(i\eta - \varepsilon)d\eta\right] \\
&= \int_{1}^{\infty} d\eta \frac{(2 - 2i)\eta^2 + (2 + 4i)\eta - 4}{\pi\eta^3\sqrt{\eta - 1}(\eta^2\tau^2 - 1)} e^{-T\eta} \\
&= \left[4\tau^2 + 2T(T + 2i) - \frac{3}{2}\right]\operatorname{erfc}\left(\sqrt{T}\right) - \frac{\sqrt{T}e^{-T}}{\sqrt{\pi}}(2T - 1 + 4i) \\
&\quad + \frac{\sqrt{\tau}e^{-T/\tau}}{\sqrt{\tau - 1}}\left[1 - i + (1 + 2i)\tau - 2\tau^2\right]\operatorname{erfc}\left(\sqrt{T/\tau}\sqrt{\tau - 1}\right) \\
&\quad + \frac{\sqrt{\tau}e^{T/\tau}}{\sqrt{\tau + 1}}\left[1 - i - (1 + 2i)\tau - 2\tau^2\right]\operatorname{erfc}\left(\sqrt{T/\tau}\sqrt{\tau + 1}\right). \tag{73}
\end{aligned}
$$

Subtracting (73) from the residue (72) gives the analytical expression in (59).

Finally, we would like to stress that the position of the pole $\gamma$ directly affects the asymptotic behavior of the final resulting function at large times $T$. The latter is determined by the smallest magnitude of $z$ encountered along $c_r$ and $c_\gamma$, i.e., by the larger of the two involved timescales: the mean-field-related 1 or the switch-on-protocol-induced $\tau$ (in units of $\tau_{\mathrm{MF}}$).

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
