# Peer review of "Out-of-equilibrium dynamical properties of Bose-Einstein condensates in a ramped up weak disorder"

_SciPost Physics_

## Round 1 · Author Response

We thank the Referee for their careful review of the manuscript and the supporting remarks for improvement. In the revised manuscript, we made a number of modifications marked in blue, which address the questions and comments of the Referee. In the following, we give a summary of changes and a point-by-point reply to the Referee. We hope that our work will be suitable for publication.
Below you will find:
i) List of changes in the amended version of manuscript No. 2501.01513v1.
ii) Reply to the comments and criticisms raised by the Referee.
Sincerely yours,
On behalf of all author

---

## Round 1 · List of Changes

1) A new paragraph at the end of Section 4 has been added.
2) Two new panels, (c) and (d), have been added to Fig.~1.
3) Panel (a) of Fig.~1 has been updated with the points showing the maximums.
4) A sentence has been added at the end of Section 5.
5) Improvements in formulation have been made.
6) The omission in the caption of Fig.~3 has been corrected.
7) The last paragraph of the Introduction has been improved.

---

## Round 2 · Author Response

Warnings issued while processing user-supplied markup:

  • Inconsistency: plain/Markdown and reStructuredText syntaxes are mixed. Markdown will be used.
    Add "#coerce:reST" or "#coerce:plain" as the first line of your text to force reStructuredText or no markup.
    You may also contact the helpdesk if the formatting is incorrect and you are unable to edit your text.

Reply to the report of Referee 2 on the manuscript No. 2501.01513v1

We thank Referee 2 for their favorable opinion and useful comments, as well as their recommendation for publication. We address each of the points mentioned by the Referee below. To make navigation easier, we marked the changes corresponding to Referee 2's criticism in magenta within the manuscript. Please note that the changes made in response to Referee 1 are still shown in blue.

Referee 2:

  1. Could the authors denote the disorder correlation function Eq. (3) and its Fourier transform in Eq. (4) by different letters?

Our response:

  1. We agree with the Referee. We changed the notation of all spatially Fourier-transformed quantities to use tildes.

Referee 2:

  1. After Eq. (28) it would be nice to explain what means small or large tau. The characteristic time scale tau_MF, to which tau should (apparently?) be compared, shows up only in Eq. (35).

Our response:

  1. Well spotted. We introduced $\tau_{\text{MF}}$ after Eq. (2) and made some clarifications accordingly after Eq. (28).

Referee 2:

  1. After Eq. (33), where the authors say that "for ... any dynamical disorder switch-on protocol" the dynamically induced part "can be at most as large as the equilibrium part". I think they mean "any tau" rather than "any protocol" since all this discussion refers to the exponential ramp Eq. (28).

Our response:

  1. We agree with the Referee. We made the necessary improvements to the text following Eq. (33).

Referee 2:

  1. According to Eqs.(43) and (49) there is an interesting dynamical threshold for tau/tau_MF=1. It can be seen mathematically (the pole c_gamma is either on the branch cut or not), but the authors give no physical interpretation of this phenomenon. Is this behavior related to the maximum of the ramping rate (short-time property) or to its long-time tail? Is it associated with the phononic spectrum and finite sound velocity (similar to the Landau criterium)? Could the authors have another look at this? I do not insist, but it would be a shame to miss something physically interesting.

Our response:

  1. This is a valuable suggestion from the Referee. Following Eq. (44) and Eq. (50), we provide a physical explanation of how the asymptotic behavior of the quantities under consideration is related to the two timescales involved: the mean field-related timescale and the switch-on protocol-induced timescale. Additionally, we briefly discuss the mathematical origin due to the role of the pole $c_\gamma$ position at the end of Appendix A.

---

## Round 2 · List of Changes

Please see the uploaded revised manuscript.
To make navigation easier, we marked the changes corresponding to Referee 2's criticism in magenta within the manuscript.
Please note that the changes made in response to Referee 1 are still shown in blue.

---

## Editorial Decision

accepted_in_target_journal